# Comparative outcomes of image-guided percutaneous catheterization versus direct visualization catheterization for peritoneal dialysis: A meta-analysis

Yi Li[‡], Lifang Li[‡], Meiju Wei, Yanxiong Qin, Yuechen Qin, Yue Zou, Haijian Zeng, Chunlan Li, Tingting Liao  *

The First Affiliated Hospital of Guangxi University of Science and Technology, Guangxi University of Science and Technology, Liuzhou, Guangxi Province, China,

‡ YL and LL are co-first authors.
* 202301405159@stdmail.gxust.edu.cn

## Abstract

### Introduction

Debate persists on the optimal catheterization method for peritoneal dialysis (PD). This meta-analysis aimed to compare the outcomes of image-guided percutaneous catheterization (IGPC) versus direct visualization catheterization (DVC) for peritoneal dialysis.

### Materials and methods

From the inception of the database until July 16, 2024, four databases (Medline, Embase, Web of Science, and the central database) were searched for literature comparing IGPC versus DVC for peritoneal dialysis. Meta-analyses were conducted on infectious complications, mechanical complications, one-year PD catheter survival, and catheter removal rates.

### Results

Totally 11 studies were included in this meta-analysis, comprising a total of 8,981 patients, of which 2,518 patients received IGPC and 6,463 patients received DVC. IGPC exhibited lower rates of infection complications (OR = 0.73, 95% CI: 0.54-0.99, P = 0.04) mechanical complications (OR = 0.64, 95% CI: 0.42-0.99, P = 0.04) and catheter removal compared to DVC (OR = 0.63, 95% CI: 0.50-0.78, P < 0.0001). However, there was no significant difference in one-year PD catheter survival between the two groups (OR = 1.33, 95% CI: 0.78-2.27, P = 0.30).

**Data availability statement:** The datasets generated/analyzed for this study can be found at the following figshare link https://figshare.com/s/9ddfc2b73777dcc78fc6.

**Funding:** The author(s) received no specific funding for this work.

**Competing interests:** The authors have declared that no ompeting interests exist.

## Conclusions

This meta-analysis concluded that IGPC was a safe and effective catheterization method for PD. The results demonstrated that IGPC significantly reduced the incidence of infection complications, mechanical complications, and catheter removal compared to DVC. No notable disparity in one-year PD survival was detected between the two groups.

## Trial registration

PROSPERO (CRD42024606795).

---

## 1 Introduction

End-stage renal disease (ESRD) is a severe kidney disease and one of the main causes of morbidity and mortality worldwide [1]. Its increasing prevalence globally poses a serious health care challenge to society. Due to the severity of this disease, there is an increasing demand for effective and economical renal replacement therapy (RRT). RRT primarily encompasses three categories: kidney transplantation, haemodialysis (HD), and peritoneal dialysis (PD). While kidney transplantation is regarded as the optimal solution, it is constrained by the shortage of donor organs and substantial medical expenses, rendering dialysis therapy the pragmatic alternative for the majority of patients. Both PD and HD have been clinically validated as viable therapy modalities. Nonetheless, PD has increasingly emerged as the favoured alternative therapy for patients with ESRD owing to its cost-effectiveness, operational simplicity, enhanced patient autonomy, suitability for home treatment, reduced dietary limitations, stable haemodynamic status, and preservation of residual renal function [2–4].

The peritoneal dialysis catheters (PDC) are essential conduits for patients to sustain dialysis treatment [5]. Infectious and mechanical problems of peritoneal catheters are the primary causes of PD failure [6]. The selection of PDC [7] and the execution of PDC operations necessitate a meticulous and cautious approach to guarantee therapeutic efficacy and safety [8,9]. This can not only effectively mitigate prevalent hazards such as infections and mechanical problems but also substantially prolong the long-term service life of PDC [10,11].

Successful PD necessitates the insertion of a PDC into the peritoneal cavity. This can be introduced using surgical methods (open or laparoscopic) or percutaneously (with or without fluoroscopy/ultrasound) [12]. Despite prior meta-analyses addressing the prognostic implications of various catheterization techniques [13,14], a consensus on the superior method remains elusive due to the distinct advantages and drawbacks inherent to each approach [15]. Open surgery and laparoscopic techniques, as direct visualization catheterization (DVC), can accurately regulate catheter positioning, with the former being the most prevalent method for putting PDC [16,17]. Open surgery offers a reduced risk of organ injury and bleeding, yet it entails greater trauma and necessitates the expertise of seasoned surgeons [18];

conversely, laparoscopic techniques present benefits like diminished trauma and expedited postoperative recovery, but are constrained by factors including equipment expenses, personnel training, and the conditions of medical institutions [19]. Certain meta-analyses indicate that the benefits of general laparoscopic catheter implantation compared to conventional open surgery are not substantial [20–22]. Nonetheless, these conclusions require further validation by large-sample, multicenter, high-quality randomised controlled trials (RCTs). Conversely, the Seldinger procedure is extensively favoured by internal medicine and interventional physicians owing to its straightforward execution and economical nature [23]. Nonetheless, there are challenges associated with imprecise positioning and heightened risk of complications during blind puncture procedures lacking imaging guidance [24]. Consequently, the execution of percutaneous methods by proficient nephrologists or interventional physicians, aided by advanced imaging guidance technologies like fluoroscopy or ultrasound, may augment their safety [25–27]. Compared to DVC, image-guided percutaneous catheterization (IGPC) provides potential benefits including a simpler process, decreased invasiveness, and the removal of delays associated with general anaesthesia and operating room scheduling [23,28–30].

Due to ongoing improvements in imaging-assisted technologies, percutaneous catheterization guided by fluoroscopy or ultrasound is receiving heightened interest. Nonetheless, There remains contention on the superiority of Scheme direct visualization catheterization or image-guided percutaneous catheterization. Therefor, we conducted a meta-analysis to compare the outcomes of direct visualization catheterization versus image-guided percutaneous catheterization for peritoneal dialysis.

## 2 Materials and methods

### 2.1 Search strategy

This meta-analysis was conducted in accordance with the Preferred Reporting Project for Systematic Review and Meta-Analysis (PRISMA) 2020 guidelines [31,32]. The databases of PubMed, Embase, Web of Science, and the Cochrane Library were systematically searched for articles from their establishment of database to July 16, 2024. The search methodology followed the PICOS principle and employed a combination of MeSH terms and unrestricted textual phrases. The search method utilised involved the amalgamation of the terms "Peritoneal Dialysis," "catheter," "direct visualization catheterization" and "image-guided percutaneous catheterization". S1 Text provides a comprehensive account of the search record.

### 2.2 Inclusion and exclusion criteria

Inclusion criteria: (1) Patients with ESRD who underwent PD catheter placement; (2) Interventional group: patients received IGPC, comprising percutaneous puncture techniques utilising X-ray fluoroscopy or ultrasound guidance; (3) Control group: patients received DVC, comprising conventional open surgery or catheter placement techniques performed under laparoscopy; (4) At least one of the following outcomes was reported: peritonitis, tunnel infection, exit-site infection, catheter dysfunction, bleeding, catheter leakage, hernia, catheter removal, and one-year PD catheter survival. (5) Study design: RCT, prospective study, or retrospective study.

Exclusion criteria: (1) Other types of articles, including conferences, abstracts, yearbooks, case reports, journals, letters, reviews, meta-analyses, editorials, pharmaceutical interventions, animal studies, and protocols; (2) Not relevant; (3) Duplicate patient cohort; (4) Inability to extract data for meta-analysis.

### 2.3 Selection of studies

The selection of articles, including the removal of duplicates, was conducted using EndNote (Version 20; Clarivate Analytics). Two reviewers conducted an initial search, independently removing duplicate entries, evaluating the titles and abstracts for relevance, and categorizing each article as either included or eliminated. The settlement was reached through consensus. The third author of the review would assume the position of arbiter in the absence of unanimity.

## 2.4 Data extraction

Two separate researchers conducted a thorough examination of the title and abstract, subsequently engaging in a comprehensive review of the entire text. A third investigator was consulted to resolve the inconsistencies. The collected data encompasses the first author's name, publication year, study area, trial ID, study design, sample size, intervention, age, study period, catheter type, infectious complications (including of peritonitis, tunnel infection, exit-site infection), mechanical complications (including of catheter dysfunction, bleeding, catheter leakage, hernia), catheter removal, and one-year PD catheter survival.

## 2.5 Risk of bias assessment

This analysis employed the Newcastle-Ottawa Scale (NOS) to evaluate the risk of bias in the included cohort studies, as assessed by two independent reviewers [33]. Two of the RCTs were assessed using the modified Jadad scale [34]. NOS comprised four topics: "research subject selection" (4 points), one item on "comparability between groups" (2 points), and three items about "outcome measurement" (3 points), with a maximum score of 9 points. Literature with six points or higher was evaluated as excellent quality. Discrepancies in the assessment revolved through group discussion regarding the contentious sections. Jadad scoring is based on the following criteria: random sequence creation, allocation concealment, blindness, and withdrawal or exit. Items 1–3 are categorised as low quality, whereas items 4–7 are designated as good grade.

## 2.6 Statistical analysis

EndNote (Version 20; Clarification Analysis) was utilised to oversee the selection of retrieved studies, including the elimination of duplicates. All study results were evaluated utilising Review Manager 5.3 (Cochrane Collaboration, Oxford, UK). The odds ratio (OR) accompanied by a 95% confidence interval (CI) was employed to compare the binary variables. Cochrane Q p-values and $I^2$ statistics were employed to assess heterogeneity in all meta-analyses. In cases of low or moderate heterogeneity ($I^2 < 50\%$), pooled data were analysed utilising a fixed effects model (FEM); conversely, if heterogeneity was significant ($\geq 50\%$), a random effects model (REM) was employed. Statistical heterogeneity was evaluated using a standard chi-square test and deemed significant at $P < 0.05$. The potential for publication bias was evaluated through a visual examination of the funnel plot. Sensitivity analyses were performed by sequentially excluding each study to assess the impact of individual studies on the aggregated results. Based on subgroup analysis conducted by directly visualizing technological differences, further assess the reliability of the research results.

# 3 Results

## 3.1 Identify eligible studies

The procedure for literature selection and inclusion is depicted in Fig 1. Our preliminary search identified a total of 971 articles. Upon integrating its data into the literature management program EndNote, 351 duplicate articles were identified and removed, resulting in a total of 620 articles. Following the assessment of the titles and abstracts, we identified and excluded 285 articles of other type, 124 articles of other diseases, and 192 articles not relevant. Subsequently, we downloaded the full texts of the remaining 19 articles and carefully read them, further excluding 3 articles failed to get full text, 3 articles failed to obtain data, and 2 articles duplicate patient cohort. Ultimately, we incorporated 11 pertinent articles. Among them, the earliest publication date was in 2008. Specific literature lists can be found in the S1 and S2 Checklist query in the Supporting Information.

## 3.2 Study characteristics

The meta-analysis encompassed 11 articles, comprising 1 prospective study, 2 RCTs, and 8 retrospective studies. A total of 8,981 cases were recorded, comprising 2,518 in the IGPC and 6,463 in the DVC group. The studies were conducted

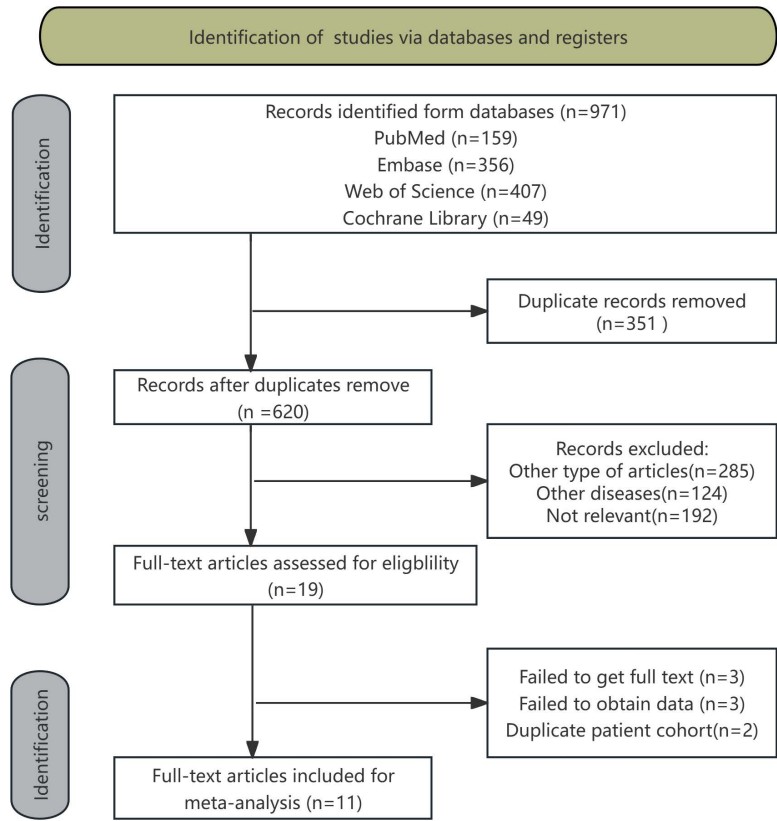

**Fig 1. Flow chart of the literature search.**

in multiple countries, including the USA, New Zealand, Brazil, and China. Table 1 presented comprehensive information and key characteristics of the patients involved in the study. Regarding the technical details of each PD catheter insertion and hospital stay, we have provided a table in S3 Text Supplementary Material 2. Details on the collection of data used for analysis and reporting can be found in S4 Dataset.

### 3.3 Risk of bias assessment

The NOS was utilised to evaluate the quality of the cohort studies. Of the nine studies, three were rated 8 points, four were rated 7 points, and two were rated 6 points. This signifies that all the incorporated cohort studies were of superior quality. The quality of the RCTs was assessed using the modified Jadad scale, and both RCTs were classified as high-quality articles. Table 1 presented the results of the risk of bias assessment, and the specific scoring process is uploaded to the S4 and S5 Dataset of the supporting information.

### 3.4 Infectious complications

Ten studies reported the occurrence of infectious complications [35–44]. The findings indicated that IGPC exhibited a reduced incidence of PD catheter-related infection complications compared to DVC, with a significant difference (OR = 0.73, 95% CI: 0.54–0.99, P = 0.04) (Fig 2).

**Table 1. General information of included studies.**

| Study | Region | study Period | Study Design | clus-ters | Sample Size (n) | Male (%) | Age, y, mean±SD | BMI, mean±SD | Catheter type | Outcomes | Quality (NOS) |
|---|---|---|---|---|---|---|---|---|---|---|---|
| [a]Rosenthal et al. 2008 [35] | Los Angeles | 1999-2004 | Retro-spective | IGPC | 54 | 44 | 56.±13.7 | NA | Tenckhoff | ①②③④⑤⑥⑦⑧ | 7 |
| | | | | DVC | 53 | 49 | 56.1±15.4 | NA | Tenckhoff | | |
| [a]Voss et al. 2012 [36] | Auckland | 1999-2004 | RCT | IGPC | 57 | 49 | 61.7±4.0 | 27±1.35 | Baxter Curl | ①②③④⑤⑥⑦⑧⑨ | 6* |
| | | | | DVC | 56 | 54 | 60.6±4.1 | 26.48±1.4 | Baxter Curl | | |
| [a]Chula et al. 2013 | Brazil | 2006-2008 | Pro-spective | IGPC | 26 | 65 | 55±15 | 25±4.34 | Tenckhoff | ④⑤⑥ | 6 |
| | | | | DVC | 42 | 40 | 60±12 | 26.5±4.87 | Tenckhoff | | |
| [a]Maher et al. 2014 [37] | New Zealand | 2004-2009 | Retro-spective | IGPC | 133 | 60 | 57.6±10.2 | 27±1.25 | Baxter Curl | ①②③④⑤⑥⑦⑧⑨ | 8 |
| | | | | DVC | 153 | 52 | 57.6±11.4 | 28.49±1.57 | Baxter Curl | | |
| [b]Sun et al. 2015 [38] | Auckland | 2009-2013 | Retro-spective | IGPC | 69 | 55 | 55.2±16.37 | 29.5±5.3 | NA | ①③⑥⑨ | 7 |
| | | | | DVC | 140 | 51 | 55.78±13.6 | 29.3±5.8 | NA | | |
| [a]Ahmed et al. 2018 | USA | 2005-2016 | Retro-spective | IGPC | 50 | 58 | 56.3±14.5 | 28.42±4.6 | NA | ①②③④⑤⑥⑦⑧⑨ | 7 |
| | | | | DVC | 190 | 41 | 52.2±17.2 | 29.05±8.22 | NA | | |
| [a]Bin Chen et al. 2021 [40] | China | 2015-2020 | RCT | IGPC | 29 | 48 | 49±11.7 | NA | Tenckhoff | ①④⑤⑥⑦ | 4* |
| | | | | DVC | 58 | 48 | 53±13.5 | NA | Tenckhoff | | |
| [a]Yibo Ma et al. 2021 [41] | China | 2017-2019 | Retro-spective | IGPC | 23 | 65 | 45.3±13.1 | 21.4±2.5 | Tenckhoff | ①②④⑥ | 7 |
| | | | | DVC | 111 | 57 | 47.6±15.1 | 22.2±3.1 | Tenckhoff | | |
| [a]Zhen Li et al. 2022 [42] | China | 2016-2019 | Retro-spective | IGPC | 105 | 57 | 49.35±14.32 | NA | Bax-ter,USA | ①②③④⑤⑥⑦⑧⑨ | 8 |
| | | | | DVC | 107 | 57 | 52.39±15.18 | NA | Bax-ter,USA | | |
| [c]Obaid et al. 2023 [43] | USA | 2017-2019 | Retro-spective | IGPC | 1781 | 49 | 60±16 | NA | NA | ③④⑥⑧ | 6 |
| | | | | DVC | 5315 | 56 | 57±18 | NA | NA | | |
| [b]Zheng et al. 2023 [44] | Northern California | 2011-2013 | Retro-spective | IGPC | 191 | 61 | 60±15.7 | 27.8±5.8 | NA | ①②③④⑤⑥⑦⑨ | 8 |
| | | | | DVC | 238 | 55 | 60.6±14.5 | 28.5±7.1 | NA | | |

Note.—IGPC: Image-guided percutaneous catheterization; DVC: direct visualization catheterization; NA: no data; ① Peritonitis; ② Tunnel infection; ③ Exit-site infection; ④ Catheter dysfunction; ⑤ Bleeding; ⑥ Catheter leak; ⑦ Hernia; ⑧ Catheter removal; ⑨ One-Year PD catheter survival. NOS: Newcas-tle–Ottawa Scale; *:Jadad:The modified Jadad scale. a:single center; b:multicenter; c:nationwide.

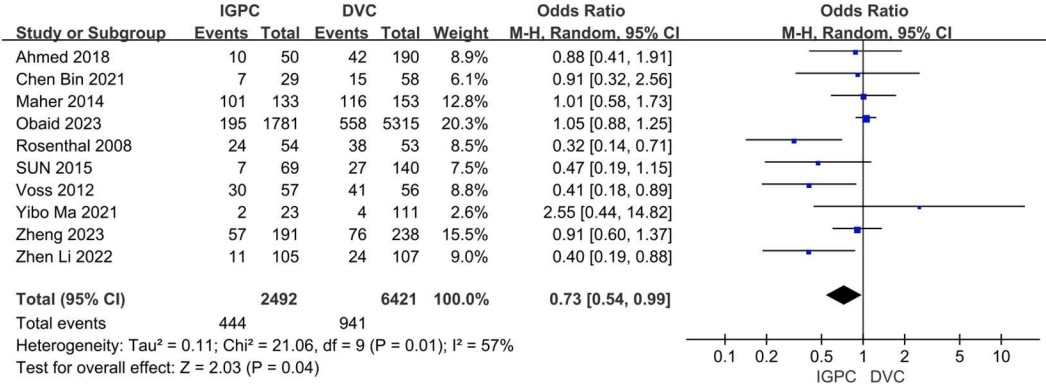

**Fig 2. Forest plot of the meta-analysis for Infection complications.**

### 3.5 Mechanical complications

Eleven studies reported the incidence of mechanical complications of PDC [35–45]. Our results indicated that the incidence of mechanical complications in the IGPC group was significantly lower compared to that in the DVC group (OR = 0.64, 95% CI: 0.42–0.99, P = 0.04) (Fig 3).

### 3.6 One-year PD catheter survival

Six studies reported the one-year survival rates of PDC [36–39,42,44]. The findings indicated that the one-year PD catheter survival in the IGPC group was marginally superior to that in the DVC group. However, the difference was not statistically significant (OR=1.33, 95% CI: 0.78–2.27, P = 0.30) (Fig 4).

### 3.7 Catheter removal

Six studies provided data regarding catheter removal [35–37,39,42,43]. Our findings indicated that the incidence catheter removal was significantly lower in the IGPC group compared to that of DVC (OR=0.63, 95% CI: 0.50–0.78, P < 0.0001) (Fig 5).

### 3.8 Sensitivity analysis

Sensitivity analysis was conducted on infectious complications, mechanical complications, one-year PD catheter survival, and catheter removal (S6 Text. Supplementary Material 3). The sensitivity analysis indicated that the results in terms of infectious

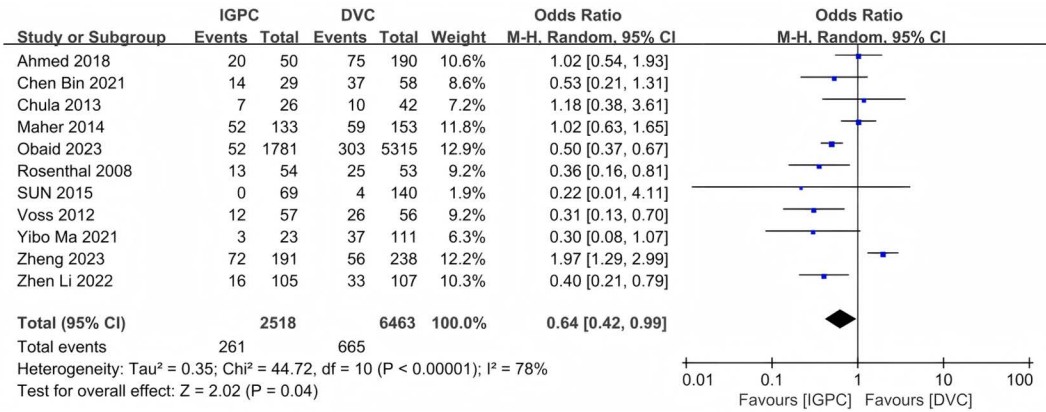

**Fig 3. Forest plot of the meta-analysis for mechanical complication.**

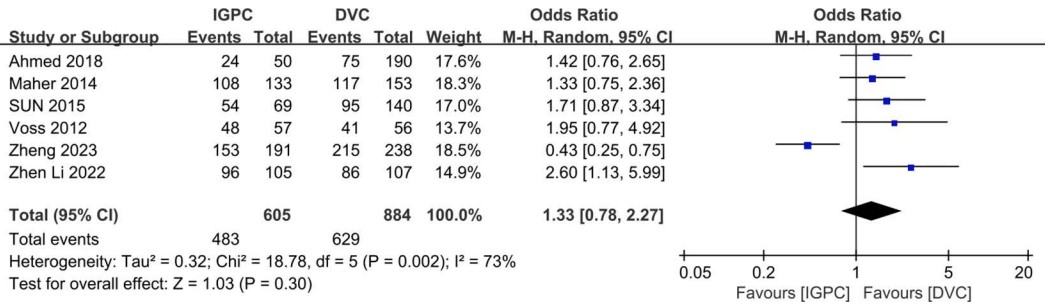

**Fig 4. Forest plot of the meta-analysis for one-year PD catheter survival.**

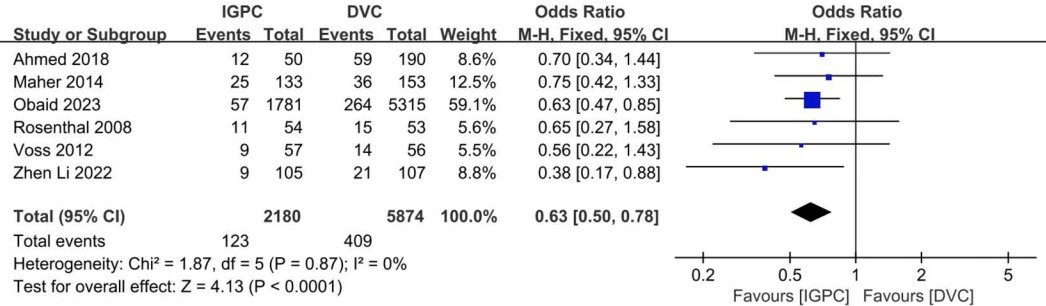

**Fig 5. Forest plot of the meta-analysis for Catheter removal.**

complications, mechanical complications and catheter removal were robust. The sensitivity analysis of the one-year PD catheter survival rate indicated that removing Zheng 2023 [44] reduced heterogeneity from 73% to 0%, and the statistical significance of the final results was altered (OR=1.63, 95%CI: 1.20–2.22, P=0.002). Upon analysing the data and sequentially eliminating the literature, the results remained unchanged, suggesting that the heterogeneity originates from this literature.

### 3.9 Subgroup analysis

Based on differences in direct visualization techniques, the 10 eligible studies were stratified into two subgroups for analysis: P vs L (IGPC group vs laparoscopic group) and P vs S (IGPC group vs open surgical group) (S7 Text. Supplementary Material 4). The results demonstrated: for infectious complications, heterogeneity was significantly reduced (I²=0%, P=0.92). Within subgroup comparisons showed comparable infection rates between the P group and both L/S groups (OR=0.88, 95% CI: 0.70–1.11, P=0.30; OR=0.86, 95% CI: 0.48–1.54, P=0.61 respectively), with no significant between-group differences. Regarding mechanical complications, heterogeneity decreased substantially (P=0.28, I²=14.0%). The P group demonstrated similar complication rates to the L group (OR=0.81, 95% CI: 0.45–1.45, P=0.47), but showed significantly lower rates compared to the S group (OR=0.53, 95% CI: 0.33–0.85, P=0.009), indicating statistically significant between-group differences. For One-year PD catheter survival, heterogeneity decreased moderately (I²=58.2%, P=0.12). Subgroup analyses revealed comparable survival rates between P group and L group (OR=1.18, 95% CI: 0.67–2.07, P=0.57), while demonstrating significantly superior survival in the P group versus S group (OR=2.60, 95% CI: 1.13–5.99, P=0.02). Catheter removal analysis showed persistent homogeneity (I²=0%, P=0.78). The P group was associated with significantly lower removal rates compared to both L and S group (OR=0.63, 95% CI: 0.50–0.80, P=0.0002; OR=0.67, 95% CI: 0.47–0.95, P=0.03).

### 3.10 Publication bias analysis

The evaluation of publication bias was conducted using funnel plots (Fig 6). The bilateral symmetric funnel plot regarding infectious complications, mechanical complications, one-year PD catheter survival, and catheter removal did not reveal any significant evidence of publication bias (Fig 6A, B, D). Conversely, the funnel plot for the one-year PD catheter survival exhibited asymmetry, suggesting a potential for publication bias (Fig 6C).

## 4 Discussion

In this meta-analysis, we systematically evaluated outcomes from studies comparing PD catheter placement via IGPC and DVC from 2000 to present. Our findings demonstrated no significant difference in one-year PD catheter technique survival between the two groups. However, IGPC was associated with a statistically significant reduction in infectious complications, mechanical complications, and catheter removal rates compared to DVC.

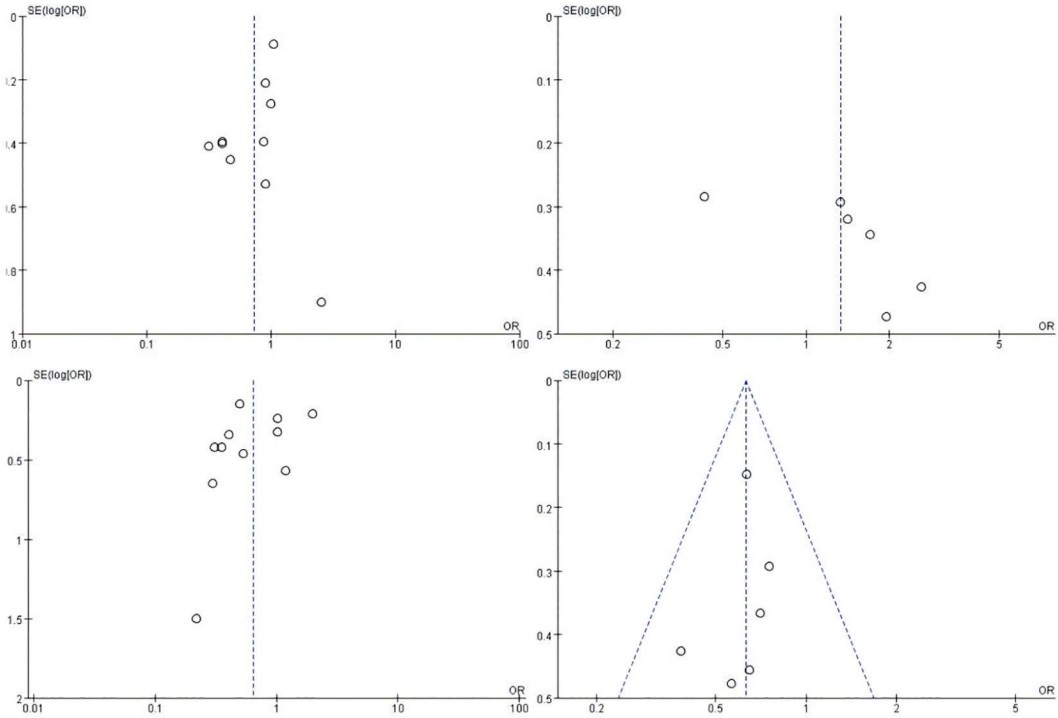

**Fig 6. Funnel plot.** (A) Infection complications. (B) Mechanical complications. (C) one-year PD catheter survival. (D) Catheter removal.

### 4.1 A general interpretation of the results in the context of other evidence

Infection is the predominant and critical complication in patients undergoing PD [46,47]. The primary causes of infection encompass dialysate leakage, inadequate aseptic practices, diminished patient immunity, chronic malnutrition, anaemia, and catheter insertion method [48]. The IGPC technique can enhance positional accuracy and effectively mitigate complications, such as infections resulting from injury to abdominal vessels, intestines, bladder, and other organs due to blind puncture or the operator's lack of experience [26].

Notably, in the largest cohort analyzed, Obaid et al. [43] reported a higher incidence of fluoroscopic-guided sepsis (4.6%, n=82) and readmission mortality (1.3%, n=24). Based on the baseline demographics and clinical characteristics of the study cohort, we observed that patients undergoing fluoroscopic intubation were more likely to be elderly, presenting with hypertension, chronic obstructive pulmonary disease (COPD), advanced liver disease without ascites, and a history of current smoking or alcohol abuse. These patients were also more likely to exhibit a significant or extreme risk of mortality.

A bidirectional pathophysiological relationship exists between infectious and mechanical complications [49]. Some studies [50–52] have mentioned highlight that peritonitis episodes may precipitate catheter dysfunction through fibrin deposition or adhesions, while mechanical complications (e.g., dialysate leaks, catheter malposition) compromise peritoneal integrity, exacerbating infection susceptibility. Our findings revealed that the incidence of mechanical complications could be reduced using the IGPC technique. A recent meta-analysis [53] suggested percutaneous techniques may lower delayed leakage rates (RR=0.35,95%

CI=0.12–0.98, $I^2$=0%) and perioperative morbidity (RR=0.25, 95% CI=0.07–0.90, $I^2$=0%), though validation through multicenter RCTs is warranted given limited existing RCT data.

Prior study [54] has indicated that minimising dressing change frequency, ensuring optimal postoperative care, and adhering to standardised fluid exchange protocols (with a focus on aseptic technique) are crucial for preventing infections at the exit site and tunnel. Patient education on aseptic technique mastery is paramount for infection prevention, as highlighted by Zhang et al. [55]. Chula et al. [45] advocated preoperative fasting (>8 hours) and antimicrobial prophylaxis (e.g., intravenous cefazolin 1g), though evidence remains inconclusive regarding perioperative antibiotic efficacy. Postoperative management protocols, Zhen Li et al. [42] advocated that patients should receive heparin saline closed catheter. On the second day, intraperitoneal injection of PD fluid (500 mL) was performed, contributed to favorable outcomes—only four IGPC patients developed catheter-related mechanical complications during one-year follow-up. Comprehensive complication management guidelines were absent in the included studies; specific details on management can be referenced in the pertinent clinical literature [56].

The one-year survival rate of PDC is a critical metric for assessing the efficacy of catheter insertion method. The ISPD advised that the minimum catheter survival rate after one year should surpass 80% [57]. Our pooled analysis revealed a comparable one-year PD catheter survival rate between IGPC and DVC.Through sensitivity analysis, we discovered that the studies conducted by Zheng et al. [44] indicated a superior one-year survival rate of IGPC in comparison to advanced laparoscopic techniques (80% vs 91%). This discrepancy might arise from variations in study design, participant demographics, as well as disparities in hospital equipment and the technical proficiency of surgeons.

In cases of catheter dysfunction, such as peritoneal catheter displacement, the catheter must be removed and a reinsertion procedure conducted, akin to tube replacement. If the peritoneal catheter is encased by the omentum and the patient's solute clearance rate is inadequate or ultrafiltration is ineffective, the catheter may be removed, or the omentum may be partially resected via laparoscopy [57]. In the included studies, IGPC for catheter removal was less frequent than the DVC method. Among them, peritonitis and patency or functional failure were the main causes of extubation, while Chula et al [45] also mentioned extubation due to patient death, transfer to HD, renal transplantation, transfer to another PD center, catheter removal through abdominal surgery, and partial recovery of renal function, without relevant data for further analysis.

### 4.2 Any limitations of the evidence included in the review

To our knowledge, this represents the first systematic meta-analysis to evaluate comparative outcomes between IGPC and DVC for peritoneal dialysis access, providing evidence to reinforce IGPC as a clinically advantageous modality. However, several methodological constraints inherent to this analysis warrant acknowledgment. First, the predominance of small sample sizes across included studies raises concerns regarding selection and publication bias, compounded by the inclusion of only two RCTs alongside a majority of single-center retrospective cohorts. The reliance on non-randomized data inherently elevates vulnerability to residual confounding, precluding robust causal inference. Notably, observational studies frequently lacked adjustment for critical confounders such as age, sex, body mass index (BMI), and prior abdominal surgical history, potentially introducing variability into effect size estimates. Furthermore, outcome-specific limitations emerged: key endpoints including one-year catheter technique survival and removal rates were reported in only six studies, resulting in underpowered subgroup analyses and possible type II error risk. The abbreviated follow-up durations across most studies further restricted the capacity to evaluate longitudinal outcomes, such as delayed mechanical complications or sustained catheter functionality. Methodologically, the aggregation of heterogeneous DVC techniques (e.g., laparoscopic vs. open surgical approaches) into a single comparator group—though pragmatically necessitated by limited data—introduced potential bias from interprocedural variability. While this stratification enabled preliminary comparative analysis, it underscores the need for future studies to delineate outcomes by specific surgical methodologies.

### 4.3 Any limitations of the review processes used

Despite exhaustive searches across four clinical trial registries, our literature search strategy may have had shortcomings, potentially leading to literature omissions and unobtainable data from completed yet unpublished studies, which could overestimate effect sizes. Some literature sources lacked key data, and despite attempts to contact authors, much data

remained unavailable. Although we performed subgroup analyses on 10 studies differentiating laparoscopic and open surgical techniques in the DVC group, results indicated non – specific potential advantages for the IGPC group across various indicators. Due to raw data limitations (e.g., absence of individual patient data), we could not conduct a planned subgroup analysis (e.g., stratified by genotype), which might obscure efficacy differences in specific populations. Additionally, we were unable to control for confounding variables such as varying inclusion criteria, population differences, and surgeon expertise levels, factors contributing to study heterogeneity and bias. Thus, more RCT – reported clinical outcomes are needed to further validate IGPC's benefits.

### 4.4  Implications of the results for practice, policy, and future research

IGPC represents a minimally invasive, cost-effective interventional radiology technique with validated safety profiles across multiple single-center investigations [58,59]. Economic evaluation data demonstrate its financial advantages: William et al. [60] reported annualized total costs of USD 69,491 for laparoscopic insertion versus USD 69,960 for open surgical placement, inclusive of postoperative care and dialysis therapy. Voss et al. [36] further confirmed its cost-effectiveness through comparative analysis, revealing laparoscopic procedures incurred nearly double the hospitalization expenses (direct costs: USD 2076 vs. USD 4125).

Notwithstanding its technical merits, the inability to perform concomitant surgical interventions during IGPC remains a significant limitation restricting real-world applicability. Conversely, this technology addresses critical clinical needs by providing viable options for patients intolerant to general anesthesia and resource-constrained primary healthcare institutions, thereby enhancing PD accessibility. Recent advancements in catheter stabilization technologies are particularly notable: Wang et al. [61] developed the "Wang clamp" device, which achieved 100% catheter tip fixation at 6-month follow-up without displacement events. Consequently, the benefits from diverse facets should be amalgamated to inform future procedural decision-making. Simultaneously, PD centres should adhere to worldwide clinical practice guidelines, consistently detect and rectify existing faults to enhance the success rate of PD catheterization.

Given the inherent limitations of this meta-analysis, it is imperative for future research to prioritize the further validation of the safety and efficacy of IGPC technology. Future research should prioritize the following: (1) Multicenter RCTs with non-inferiority designs and standardized outcome metrics; (2) Longitudinal efficacy evaluations extending beyond 12 months; (3) Comparative effectiveness studies against alternative catheterization modalities; (4) Cost-utility analyses incorporating real-world practice patterns.

## 5  Conclusion

This meta-analysis demonstrated that IGPC was both safe and effective as catheterization approach for PD. Our findings indicated that IGPC markedly diminished the occurrence of infectious complications, mechanical complications, and catheter removal in comparison to DVC. No significant difference was observed in one-year PD catheter survival between the two groups.

### Supporting information

**S1 Text.  Supplementary Material 1.**
(DOCX)

**S2 Checklist.  List of literature.**
(XLSX)

**S3 Text.  Supplementary Material 2.**
(DOCX)

**S4 Dataset. Data Extractions.**
(XLSX)

**S5 Dataset. Quality Evaluation.**
(XLSX)

**S6 Text. Supplementary Material 3.**
(DOCX)

**S7 Text. Supplementary Material 4.**
(DOCX)

**S8 File. PRISMA_2020_checklist.**
(DOCX)

## Acknowledgments

Everyone who contributed significantly to this study has been listed.

## Author contributions

**Conceptualization:** Yi Li, Lifang Li.

**Data curation:** Yi Li, Lifang Li, Meiju Wei.

**Formal analysis:** Yi Li, Lifang Li.

**Funding acquisition:** Tingting Liao.

**Investigation:** Meiju Wei, Yanxiong Qin, Yuechen Qin, Yue Zou, Chunlan Li.

**Resources:** Tingting Liao.

**Software:** Yanxiong Qin, Yuechen Qin, Yue Zou, Haijian Zeng.

**Supervision:** Yue Zou, Haijian Zeng, Chunlan Li.

**Validation:** Haijian Zeng, Chunlan Li.

**Visualization:** Tingting Liao.

**Writing – original draft:** Yi Li, Lifang Li, Meiju Wei, Yanxiong Qin, Yuechen Qin, Yue Zou, Haijian Zeng, Chunlan Li.

**Writing – review & editing:** Tingting Liao.

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
