## [Decision Letter · Decision Letter 0]

Dear Dr. Liao,

Thank you for submitting your manuscript to PLOS ONE. After careful consideration, we feel that it has merit but does not fully meet PLOS ONE’s publication criteria as it currently stands. Therefore, we invite you to submit a revised version of the manuscript that addresses the points raised during the review process.

We look forward to receiving your revised manuscript.

Kind regards,

Wisit Kaewput, MD

Academic Editor

PLOS ONE

Journal requirements: When submitting your revision, we need you to address these additional requirements. 1. Please ensure that your manuscript meets PLOS ONE's style requirements, including those for file naming. The PLOS ONE style templates can be found at https://journals.plos.org/plosone/s/file?id=wjVg/PLOSOne_formatting_sample_main_body.pdf and https://journals.plos.org/plosone/s/file?id=ba62/PLOSOne_formatting_sample_title_authors_affiliations.pdf. 2. As required by our policy on Data Availability, please ensure your manuscript or supplementary information includes the following:  A numbered table of all studies identified in the literature search, including those that were excluded from the analyses.   For every excluded study, the table should list the reason(s) for exclusion.   If any of the included studies are unpublished, include a link (URL) to the primary source or detailed information about how the content can be accessed.  A table of all data extracted from the primary research sources for the systematic review and/or meta-analysis. The table must include the following information for each study:  Name of data extractors and date of data extraction  Confirmation that the study was eligible to be included in the review.   All data extracted from each study for the reported systematic review and/or meta-analysis that would be needed to replicate your analyses.  If data or supporting information were obtained from another source (e.g. correspondence with the author of the original research article), please provide the source of data and dates on which the data/information were obtained by your research group.  If applicable for your analysis, a table showing the completed risk of bias and quality/certainty assessments for each study or outcome.  Please ensure this is provided for each domain or parameter assessed. For example, if you used the Cochrane risk-of-bias tool for randomized trials, provide answers to each of the signalling questions for each study. If you used GRADE to assess certainty of evidence, provide judgements about each of the quality of evidence factor. This should be provided for each outcome.   An explanation of how missing data were handled.   This information can be included in the main text, supplementary information, or relevant data repository. Please note that providing these underlying data is a requirement for publication in this journal, and if these data are not provided your manuscript might be rejected.   3. We note that the grant information you provided in the ‘Funding Information’ and ‘Financial Disclosure’ sections do not match.  When you resubmit, please ensure that you provide the correct grant numbers for the awards you received for your study in the ‘Funding Information’ section. 4. Thank you for stating the following financial disclosure:  [copy in funding statement].  Please state what role the funders took in the study.  If the funders had no role, please state: ""The funders had no role in study design, data collection and analysis, decision to publish, or preparation of the manuscript."" If this statement is not correct you must amend it as needed. Please include this amended Role of Funder statement in your cover letter; we will change the online submission form on your behalf.

Reviewers' comments:

Reviewer's Responses to Questions

**Comments to the Author**

1. Is the manuscript technically sound, and do the data support the conclusions?

Reviewer #1: Yes

Reviewer #2: No

Reviewer #3: Yes

Reviewer #4: Yes

Reviewer #5: Yes

Reviewer #6: Yes

Reviewer #7: Yes

2. Has the statistical analysis been performed appropriately and rigorously?

Reviewer #1: Yes

Reviewer #2: Yes

Reviewer #3: Yes

Reviewer #4: Yes

Reviewer #5: Yes

Reviewer #6: Yes

Reviewer #7: Yes

3. Have the authors made all data underlying the findings in their manuscript fully available?

Reviewer #1: Yes

Reviewer #2: Yes

Reviewer #3: Yes

Reviewer #4: Yes

Reviewer #5: Yes

Reviewer #6: Yes

Reviewer #7: Yes

4. Is the manuscript presented in an intelligible fashion and written in standard English?

Reviewer #1: Yes

Reviewer #2: Yes

Reviewer #3: Yes

Reviewer #4: Yes

Reviewer #5: Yes

Reviewer #6: Yes

Reviewer #7: Yes

Reviewer #1: This is a useful study. It will be good if there was mention about technical details used for each method of inserted the PD catheter. Perhaps a review of complications of interventional methods and their management if addressed may be useful.

Reviewer #2: Well written manuscript. Consider creating a separate section for limitations of the study and highlighting them for future studies. Also, add more updated references to the last decade 2014-2024. Thanks

Reviewer #3: Review on a manuscript titled: Comparative outcomes of image-guided percutaneous catheterization versus direct visualization catheterization for peritoneal dialysis: a meta-analysis

1. Title okay and other preliminaries,

a. Change word catheterisation to catheterization, visualisation to visualization

b. Other parts are okay, but authors may consider direct visualization as one of the key words

c. All keywords must be written in a uniform way

d. Abstract is relevant to the study

e. Conflict of interest and financial disclosure are declared

2. Introduction

a. Generally, well written and argued and the problem is apparent and concise)

b. Sentence 35 ( …. Rate of up to 100,000 per …..) need completion

c. Sentences 72-73 need revision, word therefore require completion

3. Search strategy (2.5) line 113 words ‘findings will be resolved’ should be changed to ‘revolved’

4. Analysis (2.6), well summarized, but generally difficult for non-scholars to understand, a more simplified language would make the section more understandable to most readers

5. Results (3),

a. Lines 133-136 the description of the flow chart is not clear. The description articles were sieved from 970 to 620 is clear, however it is obscure thereafter. A clear picture of what was done from 620 to 11 should be added. It is noted however the flow chart is very clear on this.

b. The rest of the result section appear well written

6. Discussion

a. Generally, well written, but

b. Paragraph one of the discussion parts appears a repetition of the introductory section than introducing the discussion

c. The discussion is generally skewed towards literature review than discussing the interpretation, limitation and implication of study findings as proposed by PRISMA checklist, authors are advised to rewrite the discussion

d. The discussion on blind insertion of the catheter in the last paragraph on infection complication of the discussion (230-232) is out of scope of this study. This review was on image guided peritoneal catheter insertion vs direct vision (laparoscopic and open).

e. Discussion on mechanical complications lines 234-239 too on based on a topic which was reviewed in the current meta-analysis. Authors are advised to adhere to PRISMA checklist

f. On the discussion on catheter removal line 252, there a typo (catheter ing) which need to be corrected,

g. Line 262 words ‘markedly inferior to’ may be replaced by words ‘less frequent ..’

h. Line 263 last sentence ‘IGPC demonstrated the advantage of visualization, resulting in decreased rates of infection and mechanical problems’ is misleading and may not be originating from this review

i. The study limitations at the end of the discussion

7. Conclusion

a. Well written, but hidden within the discussion texts

b. Conclusion is informed by review findings

8. Funding – declared as required, authors should be put word ‘and’ before the last sentence

9. References and attachments

a. References are according to journal, but reference 17 may not be relevant

b. All attachments are available

Reviewer #4: In this study, the author conducted metaanalysis to comapre image-guided percutaneous insertion or surgical (open or laparoscopic) insertion of peritoneal dialysis catheter. They found image-guided percutaneous placement had lower rate of infectious and mechanical complications. catheter removal rate was low in percutaneous group as well. they concluded percutaneous insertion is a safe and effective procedure.

The authors mixed open and laparoscopic procedure to a single group, but I wonder some comlication such as wound infection might be significantly different between laparoscopic and open surgery.

What is the difference of this meta-analysis compared to previous one (DOI: 10.1177/20543581211052731) and what kind of new information is added? Because several meta-analysis have been published recently, I reccomend authors clarify why they permorm the same comparison.

Why were some studies included previous metaanalysis (i.e. doi: 10.3747/pdi.2013.00003, DOI: 10.1007/s40620-020-00896-w) not included in this study?

minor point

line 133)

Following the assessmentof the titles and abstracts, 19 articles were deemed inappropriate and subsequently excluded. Ultimately, we incorporated 11 pertinent articles.

Number doesn't match the figure 1. probably 8 instead of 19?

line 163, 185)

Mechanical complications -> mechanical complications

Reviewer #5: Dear Authors

congratulation for your hard work and meta-analysis to compere the two common techniques for peritoneal dialysis and support the image-guided percutaneous catheterisation (IGPC) and direct visualization catheterisation(DVC) and conclude the superiority of IGPC for PD.

thank you very much

Reviewer #6: The authors present an interesting study looking at a meta analysis between image-guided PD catheter insertion vs. direct catheter insertion. The authors show that IGPD had lower infection rates, mechanical complications, and catheter removal. There was no difference noted in 1-year PD catheter survival between the 2 cohorts.

The authors state that the articles included were most recently up to 7/16/24. However, the authors did not include when the earliest date of the studies included in this meta analysis was published and would be very helpful in interpreting these results. Obviously, over time, our perioperative infectious rates have improved with antibiotic and sterilization technique improvements. In addition, technological improvements in the development of PD catheters has also significantly improved since their inception.

What are the credentials of the 2-3 investigators that performed this analysis? Do they hold professional degrees? Are they clinically oriented or purely research oriented personnel?

Are the authors advocating that all PD catheters are inserted under image-guidance? What is the cost difference of utilizing image-guidance vs. direct insertion? This data would really help strengthen the authors' already strong argument.

Did the direct visualization cohort include MIS and open tecniques? Was there any difference between direct visualization techniques themselves? A sub group analysis of differences between direct. visualization techniques would also help improve the impact of this study.

Reviewer #7: Peritoneal dialysis (PD) is an importend treatment modality for end stage renal patients.

The Paper is a meta-analysis looking at the optimal catheterisation method for peritoneal

dialysis (PD), comparing the outcomes of image-guided percutaneous catheterisation (IGPC) versus direct visualisation catheterisation (DVC).

It includes 11 studies from multiple countries, comprising a total of 8,981 patients, of which 2,518 patients received IGPC and 6,463 patients received DVC.

The high number of patients originates from 1 American article with 7096 patients, while the others have much lower numbers.

They looked at infectious complications, mechanical complications, one-year PD catheter survival, and catheter removal and found no significant disparity regarding one-year PD catheter survival between two groups. Nonetheless, IGPC significantly reduced the incidence of infectious complications, mechanical complications and catheter removal compared with DVC.

The limitations are discussed.

Comments:

Table 1:

Gender should also be written in percentages for easier comparison.

It should be noted if single center or multicenter /or nationwide.

BMI should be decided by group (IGPC/DVC)

Cathetertype – it should be noted if it is a one or two- cuff catheter

It should be noted in more detail how the procedures are done in each study, including a decription of how long the patient had to be in the hospital for the procedure.

Cathter removal can also be due to the patient being transplanted, it is not addressed if this is included as a cause for removal

The authors conclude it is safe to use IGPC and also that the infectious complications were reduced. In the largest of the included studies, they state that IGPC had had higher sepsis and mortality – this should be discussed.

**Do you want your identity to be public for this peer review?** For information about this choice, including consent withdrawal, please see our Privacy Policy

Reviewer #1: No

Reviewer #2: **Yes: ** Mahmoud Elfiky

Reviewer #3: **Yes: ** Masumbuko Y. Mwashambwa

Reviewer #4: **Yes: ** Katsuhiro Ito

Reviewer #5: **Yes: ** Abdullah Saeed Abdullah

Reviewer #6: No

Reviewer #7: No

---

## [Author Response · Author response to Decision Letter 1]

17 Mar 2025

Response: We ensure that our manuscript meets PLOS ONE's style requirements.

2. As required by our policy on Data Availability, please ensure your manuscript or supplementary information includes the following: 

Response: All of this information has been submitted within the main text and uploaded in the form of supplementary information. All studies obtained from literature searches: S 2 Checklist; A summary table of all data extracted from the main sources of research: S 4 Dataset; Bias risk and quality/certainty assessment: S 5 Dataset.

 Response: Thank you for the reminder. Upon reconfirmation, we have found that we have not received any financial support. The text has been modified accordingly. Please check the revised lines 340-341.

4.Thank you for stating the following financial disclosure: 

 [copy in funding statement].  

Response: Thank you for your correction. We have added an amended Role of Funder statement to our cover letter. Please assist in updating the form submitted online. We also wish to add something: due to our negligence, we discovered that in the initial submission of information, we mistakenly wrote the corresponding author's information as "Guangxi Medical University First Affiliated Hospital: The First Affiliated Hospital of Guangxi Medical University" instead of "The First Affiliated Hospital of Guangxi University of Science and Technology, Guangxi University of Science and Technology". This was an error, and it has been corrected upon resubmission.

Reviewer #1: 

This is a useful study. It will be good if there was mention about technical details used for each method of inserted the PD catheter. Perhaps a review of complications of interventional methods and their management if addressed may be useful.

Response: Thank you for your affirmation of our research. Due to the excessive length, regarding the technical details of each PD catheter insertion and hospital stay, we have provided a table in S3 Text Supplementary Material 2. We reviewed the prevention and management of complications related to the interventional method in lines 255-266 of the text.

Reviewer #2: 

Well written manuscript. Consider creating a separate section for limitations of the study and highlighting them for future studies. Also, add more updated references to the last decade 2014-2024. Thanks

Response: Thank you for your suggestion. We have discussed the limitations of our study in lines 281-311 and updated with references added from the past decade.

Reviewer #3:

Review on a manuscript titled: Comparative outcomes of image-guided percutaneous catheterization versus direct visualization catheterization for peritoneal dialysis: a meta-analysis

1. Title okay and other preliminaries,

a. Change word catheterisation to catheterization, visualisation to visualization

b. Other parts are okay, but authors may consider direct visualization as one of the key words

c. All keywords must be written in a uniform way

d. Abstract is relevant to the study

e. Conflict of interest and financial disclosure are declared

Response: Regarding spelling errors: Thank you for your careful examination. We apologize for our carelessness. Based on your suggestions, we have made modifications to ensure the consistency of the vocabulary throughout the entire manuscript.

2. Introduction

a. Generally, well written and argued and the problem is apparent and concise)

b. Sentence 35 ( …. Rate of up to 100,000 per …..) need completion

c. Sentences 72-73 need revision, word therefore require completion

Response: Modified.

3. Search strategy (2.5) line 113 words ‘findings will be resolved’ should be changed to ‘revolved’

Response: Modified.

4. Analysis (2.6), well summarized, but generally difficult for non-scholars to understand, a more simplified language would make the section more understandable to most readers

Response: Your consideration is meaningful, but it belongs to the operational basis in our research process, which has been written in a standardized manner and cannot be further simplified.

5. Results (3),

a. Lines 133-136 the description of the flow chart is not clear. The description articles were sieved from 970 to 620 is clear, however it is obscure thereafter. A clear picture of what was done from 620 to 11 should be added. It is noted however the flow chart is very clear on this.

b. The rest of the result section appear well written

Response: We have supplemented and improved on lines 139-142.

6. Discussion

a. Generally, well written, but

b. Paragraph one of the discussion parts appears a repetition of the introductory section than introducing the discussion

c. The discussion is generally skewed towards literature review than discussing the interpretation, limitation and implication of study findings as proposed by PRISMA checklist, authors are advised to rewrite the discussion

d. The discussion on blind insertion of the catheter in the last paragraph on infection complication of the discussion (230-232) is out of scope of this study. This review was on image guided peritoneal catheter insertion vs direct vision (laparoscopic and open).

e. Discussion on mechanical complications lines 234-239 too on based on a topic which was reviewed in the current meta-analysis. Authors are advised to adhere to PRISMA checklist

f. On the discussion on catheter removal line 252, there a typo (catheter ing) which need to be corrected,

g. Line 262 words ‘markedly inferior to’ may be replaced by words ‘less frequent ..’

h. Line 263 last sentence ‘IGPC demonstrated the advantage of visualization, resulting in decreased rates of infection and mechanical problems’ is misleading and may not be originating from this review

i. The study limitations at the end of the discussion

Response: Thank you for your feedback. After discussion, we have structured our discussion content into four subheadings based on the discussion section of the PRISMA checklist and have reorganized our discussion accordingly. Please refer to the revised lines 228-334.

7. Conclusion

a. Well written, but hidden within the discussion texts

b. Conclusion is informed by review findings

Response: Modified.

8. Funding – declared as required, authors should be put word ‘and’ before the last sentence

Response: Added.

9. References and attachments

a. References are according to journal, but reference 17 may not be relevant

b. All attachments are available

Response: Thank you for promptly correcting our error, we have replaced the document, please refer to the revised lines 411-413.

Reviewer #4:

 In this study, the author conducted metaanalysis to comapre image-guided percutaneous insertion or surgical (open or laparoscopic) insertion of peritoneal dialysis catheter. They found image-guided percutaneous placement had lower rate of infectious and mechanical complications. catheter removal rate was low in percutaneous group as well. they concluded percutaneous insertion is a safe and effective procedure.

The authors mixed open and laparoscopic procedure to a single group, but I wonder some comlication such as wound infection might be significantly different between laparoscopic and open surgery.

Response: Your consideration is necessary. We found in a CAPD I Trial (DOI: 10.3747/pdi.2017.00023) that compared the outcomes of open placement with laparoscopic placement of PD catheters, the experiment indicated that the clinical success rates of both techniques are the same. In another randomized controlled trial (https://doi.org/10.1186/s12882-020-01724-w), laparoscopic PD catheter insertion may be superior to conventional open catheter insertion, but its bleeding incidence (OR: 3.25, 95% CI: 1.18 to 8.97, P: 0.02) is higher than the open surgical catheter insertion technique. To further confirm our conclusion, we also conducted a subgroup analysis of each technique (advanced image-guided percutaneous puncture, laparoscopy, open surgery), and the results still showed that the IGPC group has potential advantages. However, this conclusion needs to be confirmed with further large-sample-size, multi-center, high-quality RCTs.

What is the difference of this meta-analysis compared to previous one (DOI: 10.1177/20543581211052731) and what kind of new information is added? Because several meta-analysis have been published recently, I reccomend authors clarify why they permorm the same comparison.

Why were some studies included previous metaanalysis (i.e. doi: 10.3747/pdi.2013.00003, DOI: 10.1007/s40620-020-00896-w) not included in this study?

Response: In our study, it is explicitly required that the percutaneous group involves the placement of peritoneal dialysis catheters under the guidance of fluoroscopy or ultrasound. To our knowledge, this is also the first meta-analysis comparing image-guided percutaneous puncture techniques with direct visualization techniques. However, the inclusion criteria for Percutaneous Versus Surgical Insertion of Peritoneal Dialysis Catheters: A Systematic Review and Meta-Analysis (DOI: 10.1177/20543581211052731) did not exclude studies using blinding methods. Similarly, some studies included in previous meta-analyses (e.g., doi: 10.3747/pdi.2013.00003, DOI: 10.1007/s40620-020-00896-w) also included blinding in their percutaneous group, which does not meet our inclusion criteria.

minor point

line 133)

Following the assessmentof the titles and abstracts, 19 articles were deemed inappropriate and subsequently excluded. Ultimately, we incorporated 11 pertinent articles.

Number doesn't match the figure 1. probably 8 instead of 19?

line 163, 185)

Mechanical complications -> mechanical complications

Response: Thank you for your correction, it has been modified.

Reviewer #5:

Dear Authors

congratulation for your hard work and meta-analysis to compere the two common techniques for peritoneal dialysis and support the image-guided percutaneous catheterisation (IGPC) and direct visualization catheterisation(DVC) and conclude the superiority of IGPC for PD.

thank you very much

Response: Thank you for your affirmation.

Reviewer #6: 

The authors present an interesting study looking at a meta analysis between image-guided PD catheter insertion vs. direct catheter insertion. The authors show that IGPD had lower infection rates, mechanical complications, and catheter removal. There was no difference noted in 1-year PD catheter survival between the 2 cohorts.

The authors state that the articles included were most recently up to 7/16/24. However, the authors did not include when the earliest date of the studies included in this meta analysis was published and would be very helpful in interpreting these results. Obviously, over time, our perioperative infectious rates have improved with antibiotic and sterilization technique improvements. In addition, technological improvements in the development of PD catheters has also significantly improved since their inception.

Response: Thank you for your question. Among us, one is a Doctor of Clinical Medicine, Associate Researcher; the other two are Bachelor of Clinical Medicine, Clinical Doctors.

What are the credentials of the 2-3 investigators that performed this analysis? Do they hold professional degrees? Are they clinically oriented or purely research oriented personnel?

Response: Thank you for your question. Among us, one is a Doctor of Clinical Medicine, Associate Researcher; the other two are Bachelor of Clinical Medicine, Clinical Doctors.

Are the authors advocating that all PD catheters are inserted under image-guidance?

Response: Although percutaneous catheter placement guided by imaging has its advantages, for those who require complex surgical interventions, such as hernia repair and adhesiolysis, surgical placement of PD catheters, whether laparoscopic or open, we believe should be the preferred surgical method. Overall, the decision on the placement of PD catheters should be individualized and made by a multidisciplinary team consisting of nephrologists, surgeons, interventional radiologists, and anesthesiologists.

What is the cost difference of utilizing image-guidance vs. direct insertion? This data would really help strengthen the authors' already strong argument.

Response: Thank you for your suggestion, we have consulted the relevant literature and discussed the matter on lines 313-319.

Did the direct visualization cohort include MIS and open tecniques? Was there any difference between direct visualization techniques themselves?

Response: Yes, the direct visualization queue includes MIS and open technologies, and their differences are discussed in the introduction section (please refer to the revised lines 56-63).

A sub group analysis of differences between direct. visualization techniques would also help improve the impact of this study.

Response: Thank you for your feedback, we have conducted subgroup analysis based on your suggestion (please refer to the revised lines 201-217).

Reviewer #7: 

Peritoneal dialysis (PD) is an importend treatment modality for end stage renal patients.

The Paper is a meta-analysis looking at the optimal catheterisation method for peritoneal

dialysis (PD), comparing the outcomes of image-guided percutaneous catheterisation (IGPC) versus direct visualisation

---

## [Decision Letter · Decision Letter 1]

Dear Dr. Liao,

Thank you for submitting your manuscript to PLOS ONE. After careful consideration, we feel that it has merit but does not fully meet PLOS ONE’s publication criteria as it currently stands. Therefore, we invite you to submit a revised version of the manuscript that addresses the points raised during the review process.

We look forward to receiving your revised manuscript.

Kind regards,

Wisit Kaewput, MD

Academic Editor

PLOS ONE

Journal Requirements:

Reviewers' comments:

Reviewer's Responses to Questions

**Comments to the Author**

Reviewer #1: All comments have been addressed

Reviewer #2: All comments have been addressed

Reviewer #4: All comments have been addressed

Reviewer #6: (No Response)

Reviewer #7: All comments have been addressed

2. Is the manuscript technically sound, and do the data support the conclusions?

Reviewer #1: Yes

Reviewer #2: Yes

Reviewer #4: Yes

Reviewer #6: (No Response)

Reviewer #7: Yes

3. Has the statistical analysis been performed appropriately and rigorously?

Reviewer #1: Yes

Reviewer #2: Yes

Reviewer #4: Yes

Reviewer #6: (No Response)

Reviewer #7: N/A

4. Have the authors made all data underlying the findings in their manuscript fully available?

Reviewer #1: Yes

Reviewer #2: Yes

Reviewer #4: Yes

Reviewer #6: (No Response)

Reviewer #7: Yes

5. Is the manuscript presented in an intelligible fashion and written in standard English?

Reviewer #1: Yes

Reviewer #2: Yes

Reviewer #4: Yes

Reviewer #6: (No Response)

Reviewer #7: Yes

Reviewer #1: (No Response)

Reviewer #2: Thank you for addressing all concerns. This study can be the pilot for more upcoming studies to address the issue

Reviewer #4: (No Response)

Reviewer #6: The authors have answered almost all of my comments. There is still one questions in which they have no addressed yet:

"The authors state that the articles included were most recently up to 7/16/24. However, the authors did not

include when the earliest date of the studies included in this meta analysis was published and would be

very helpful in interpreting these results. Obviously, over time, our perioperative infectious rates have

improved with antibiotic and sterilization technique improvements. In addition, technological

improvements in the development of PD catheters has also significantly improved since their inception."

I think inclusion of how early these studies have started will really help inform the reader about the outcomes seen in this study.

Reviewer #7: Thank you for resubmitting this manuscript and addressing the comments from all reviewers.

**Do you want your identity to be public for this peer review?** For information about this choice, including consent withdrawal, please see our Privacy Policy

Reviewer #1: **Yes: ** ashish jiwane

Reviewer #2: **Yes: ** Mahmoud Elfiky

Reviewer #4: **Yes: ** Katsuhiro Ito

Reviewer #6: No

Reviewer #7: No

---

## [Author Response · Author response to Decision Letter 2]

10 May 2025

If applicable, we recommend that you deposit your laboratory protocols in protocols.io to enhance the reproducibility of your results. Protocols.io assigns your protocol its own identifier (DOI) so that it can be cited independently in the future. For instructions see: https://journals.plos.org/plosone/s/submission-guidelines#loc-laboratory-protocols.

Response: First of all, thank you for your suggestion. We have already registered the study protocol on PROSPERO (number CRD42024606795), so we do not intend to store the lab protocol in protocols.io anymore.

Reviewer #6: The authors have answered almost all of my comments. There is still one question which they have not addressed yet:

"The authors state that the articles included were most recently up to 7/16/24. However, the authors did not include when the earliest date of the studies included in this meta-analysis was published, which would be very helpful in interpreting these results. Obviously, over time, our perioperative infectious rates have improved with antibiotic and sterilisation technique improvements. In addition, technological improvements in the development of PD catheters have also significantly improved since their inception."

I think inclusion of how early these studies have started will really help inform the reader about the outcomes seen in this study.

Response: Thank you for the reminder that this was an oversight on our part. The issue has been modified in the original text, as detailed on pages 17, 82-83, and 143-144 of the revised text.

---

## [Editor Report · Decision Letter 2]

Comparative outcomes of image-guided percutaneous catheterization versus direct visualization catheterization for peritoneal dialysis: a meta-analysis

PONE-D-24-51344R2

Dear Dr. Liao,

We’re pleased to inform you that your manuscript has been judged scientifically suitable for publication and will be formally accepted for publication once it meets all outstanding technical requirements.

Kind regards,

Wisit Kaewput, MD

Academic Editor

PLOS ONE

Additional Editor Comments (optional):

Accept as is.
---

## [Editor Report · Acceptance letter]

PONE-D-24-51344R2

PLOS ONE

Dear Dr. Liao,

I'm pleased to inform you that your manuscript has been deemed suitable for publication in PLOS ONE. Congratulations! Your manuscript is now being handed over to our production team.

Kind regards,

on behalf of

Dr. Wisit Kaewput

Academic Editor

PLOS ONE